# Clinical, financial and social impacts of COVID-19 and their associations with mental health for mothers and children experiencing adversity in Australia

Hannah Bryson[1,2], Fiona Mensah[2,3], Anna Price[1,2,3], Lisa Gold[4], Shalika Bohingamu Mudiyanselage[4], Bridget Kenny[1,2], Penelope Dakin[5], Tracey Bruce[6], Kristy Noble[5], Lynn Kemp[6], Sharon Goldfeld[1,2,3]*

1 Centre for Community Child Health, The Royal Children's Hospital, Parkville, VIC, Australia, 2 Population Health, Murdoch Children's Research Institute, Parkville, VIC, Australia, 3 Department of Paediatrics, University of Melbourne, Parkville, VIC, Australia, 4 School of Health and Social Development, Deakin University, Burwood, VIC, Australia, 5 Australian Research Alliance for Children and Youth, Canberra City, ACT, Australia, 6 Ingham Institute, Western Sydney University, NSW, Australia

* sharon.goldfeld@rch.org.au

**Data Availability Statement:** Data cannot be shared publicly because they contain sensitive

## Abstract

### Background

Australia has maintained low rates of SARS-COV-2 (COVID-19) infection, due to geographic location and strict public health restrictions. However, the financial and social impacts of these restrictions can negatively affect parents' and children's mental health. In an existing cohort of mothers recruited for their experience of adversity, this study examined: 1) families' experiences of the COVID-19 pandemic and public health restrictions in terms of clinical exposure, financial hardship family stress, and family resilience (termed 'COVID-19 impacts'); and 2) associations between COVID-19 impacts and maternal and child mental health.

### Methods

Participants were mothers recruited during pregnancy (2013–14) across two Australian states (Victoria and Tasmania) for the 'right@home' trial. A COVID-19 survey was conducted from May-December 2020, when children were 5.9–7.2 years old. Mothers reported COVID-19 impacts, their own mental health (Depression, Anxiety, Stress Scales short-form) and their child's mental health (CoRonaVIruS Health and Impact Survey subscale). Associations between COVID-19 impacts and mental health were examined using regression models controlling for pre-COVID-19 characteristics.

### Results

319/406 (79%) mothers completed the COVID-19 survey. Only one reported having had COVID-19. Rates of self-quarantine (20%), job or income loss (27%) and family stress (e.g., difficulty managing children's at-home learning (40%)) were high. Many mothers also

participant information which are restricted to use for research purposes only. This restriction is in accordance with the Participant Information and Consent Form and approved study protocol, governed by the Royal Children's Hospital Human Research Ethics Committee (HREC 32296). We invite researchers to request access to the data from the Melbourne Children's Campus LifeCourse institutional data access platform (https://lifecourse.melbournechildrens.com/data-access/) or the governing Royal Children's Hospital HREC.

**Funding:** "right@home" is funded by the Victorian Department of Education and Training (https://www.education.vic.gov.au/), the Tasmanian Department of Health and Human Services (https://www.health.tas.gov.au/, the Ian Potter Foundation (https://www.ianpotter.org.au/), Sabemo Trust, Sidney Myer Fund (https://www.myerfoundation.org.au/), the Vincent Fairfax Family Foundation (https://vfff.org.au/) and the Australian National Health and Medical Research Council (NHMRC, Project Grant 1079148 - https://www.nhmrc.gov.au/). The right@home COVID-19 data collection was supported by the Victorian Department of Health and Human Services (https://www.dhhs.vic.gov.au/), Morgan Stanley (https://www.morganstanley.com.au/) and the Vincent Chiodo Charitable Trust. Research at the MCRI is supported by the Victorian Government's Operational Infrastructure Support Program. FM is supported by NHMRC Career Development Fellowship 1111160. SG is supported by NHMRC Practitioner Fellowship 1155290. The funders had no role in study design, data collection and analysis, decision to publish, or preparation of the manuscript.

**Competing interests:** The authors have declared that no competing interests exist.

reported family resilience (e.g., family found good ways of coping (49%)). COVID-19 impacts associated with poorer mental health (standardised coefficients) included self-quarantine (mother: $\beta$ = 0.46, child: $\beta$ = 0.46), financial hardship (mother: $\beta$ = 0.27, child: $\beta$ = 0.37) and family stress (mother: $\beta$ = 0.49, child: $\beta$ = 0.74). Family resilience was associated with better mental health (mother: $\beta$ = -0.40, child: $\beta$ = -0.46).

## Conclusions

The financial and social impacts of Australia's public health restrictions have substantially affected families experiencing adversity, and their mental health. These impacts are likely to exacerbate inequities arising from adversity. To recover from COVID-19, policy investment should include income support and universal access to family health services.

## Introduction

The coronavirus SARS-COV-2 (COVID-19) was first confirmed in Australia in January 2020. Strict public health restrictions ("lockdown") were imposed nationally from March 2020 to reduce the spread of the virus (see Fig 1). As of 31 December 2020, Australia had recorded an overall incidence rate of 111 cases and 3.5 deaths per 100,000 people [1]. The peak of infections occurred in July 2020 when the State of Victoria experienced a second wave of infections, with the national rate reaching 721 new cases (2.8 per 100,000) in 24 hours. These figures are relatively low compared with other high-income countries such as the United States (US) and United Kingdom (UK), which had recorded overall rates of 5895 and 3730 cases per 100,000 respectively, with peaks at around 70 to 75 new cases in 24 hours per 100,000 [1]. While COVID-19 symptoms are generally less severe in children and young adults, [2–4] the significant economic and social impacts have negatively affected the mental health of parents and children, both in Australia [5, 6] and internationally [7, 8]. Emerging global evidence suggests that these impacts disproportionately affect families who were experiencing adversity (e.g. parental unemployment, low educational attainment, relationship difficulties, poor mental health) before the pandemic [9, 10]. However, there are few empirical studies examining the

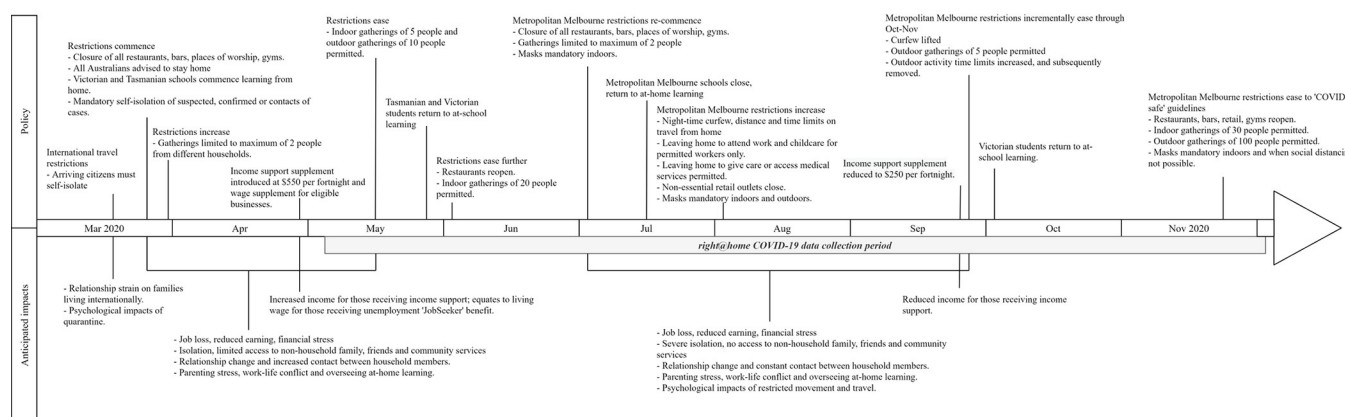

**Fig 1. Timeline of COVID-19 policy implementation and impacts on families.**

impacts of the COVID-19 pandemic for mothers and children who were already experiencing adversity before the pandemic.

Fig 1 describes the public health restrictions implemented by the Australian Federal and State Governments across the first 10 months of the pandemic [11–13]. During the national lockdown from March to June 2020, all Australians were advised to stay home, schools were temporarily closed, children transitioned to at-home learning, limits were placed on visitors in the home and operation of non-essential businesses was restricted. As lockdown eased for all States and Territories, Metropolitan Melbourne was uniquely impacted by a second lockdown with additional restrictions implemented from July to November 2020 [11]. These included gatherings limited to two people, night-time curfews, limits on distance (5 kilometres) time (1 hour) and reasons for leaving the home (essential services, exercise and providing care), and restricted access to early childhood education and care providers.

Australia's lockdown measures led to widespread job and income loss. Between March and July 2020, Australian unemployment increased from 5.2% to 7.5% (a 20-year high) and under-employment from 8.8% to 11.7% [14]. To mitigate these economic impacts, the Australian Government implemented policies which included a Coronavirus supplement to unemployment benefits ('JobSeeker') which effectively doubled the existing benefit from $550 to $1,100 a fortnight [15]. A wage subsidy was also introduced for eligible businesses to support retention of their workforce ('JobKeeper'), and free childcare was implemented for working families [16]. However, the amount, eligibility and duration of these benefits were progressively reduced through the latter half of 2020, with free childcare ended by July 2020 and both the Coronavirus supplement and JobKeeper ended by April 2021 [15, 16]. Despite these policies, many families have experienced substantial financial stress and hardship, [6, 17] with emerging evidence that young adults and women were among those most impacted [18].

The nexus between financial hardship, such as parent unemployment and loss of income, and poorer child and family mental health is well researched [19]. Financial hardship impacts parents' ability to pay for essentials and to care for and invest in resources for their children. Additionally, it puts families at greater risk of poor mental health, family violence and child maltreatment [20–22]. Subsequently, poor parent mental health can negatively impact parents' own health, [23] as well as their children's behavioural and emotional outcomes [24, 25]. Research from previous pandemics has demonstrated the negative psychological impacts of quarantine, social isolation and loss of schooling on child and family mental health outcomes such as depression, anxiety, stress, anger and confusion [26, 27].

Research examining the impacts of the COVID-19 pandemic and public health restrictions suggests mental health difficulties amongst parents and children have increased [5, 28, 29]. For example, a national survey of Australian households (the National Child Health Poll) in June 2020 found almost half of parents (48%) reported that the pandemic had negatively impacted their mental health, and this was more likely amongst those who had experienced financial impacts [6]. Similarly, a nationally weighted survey of parents in the United States in June 2020 found that 27% of parents reported their mental health had declined during the first 3 months of the pandemic [30]. Families reporting poorer mental health were also more likely to have lost access to regular childcare and be experiencing financial hardship (assessed as food insecurity). These findings were again mirrored by those of a UK study which surveyed families in a highly deprived and ethnically diverse city in April to June 2020; this study found high rates of poor mental health amongst mothers (39–43%) and that poor mental health was also associated with unemployment, housing insecurity and lack of social support during the public health restrictions [7].

With regard to children's mental health, an online survey of 5823 parents in Austria, Germany Liechtenstein and Switzerland in April to May 2020 found that 33–43% of parents

reported an increase in their child's emotional and behavioural difficulties from the onset of the pandemic, amongst 1- to 6-year-olds [31]. Similarly, a nationally representative study in Germany which surveyed families between May to June 2020 found that rates of mental health difficulties had approximately doubled from pre-pandemic figures, amongst 7- to 17-year-olds. Again, these increases in poor mental health were greatest amongst families experiencing socioeconomic adversity and were also greater amongst younger children [32]. For families who were experiencing adversity before the pandemic, limited resources and supports are likely to be further diminished by both the financial and social effects of the pandemic and related restrictions [10]. These impacts are expected to increase the risk of poor parent and child mental health amongst families experiencing adversity, with children bearing a disproportionate burden.

To be effective and equitable, Australia's public policy responses for post-COVID-19 recovery must consider how lockdown has affected families experiencing adversity; including both risk and protective factors that have emerged. To this end, the current study uses data collected opportunistically during the first 10 months of the COVID-19 pandemic in Australia, in an established cohort of mothers and children who were recruited for their experience of adversity. Mothers were recruited during pregnancy in two states of Australia (Victoria and Tasmania) as part of the 'right@home' trial [33]; their children were 5 to 7 years old when both lockdown periods occurred between March and December 2020. In this cohort of Australian mothers and children, our study aimed to examine: 1) families' experiences of the COVID-19 pandemic and public health restrictions in terms of clinical exposure, changes to financial circumstances, current financial hardship, family stress and family resilience (termed 'COVID-19 impacts'); and 2) the associations between these COVID-19 impacts and maternal and child mental health.

## Methods

### Design and setting

This was an Australian cohort study drawing on data collected within the 'right@home' randomised controlled trial of nurse home visiting. In 2013–14, 722 women were recruited to the trial based on their experience of adversity during pregnancy (summarized below and published in detail), at which time they provided written informed consent to participate [33]. The nurse home visiting program (the intervention) was embedded into the existing universal child and family health service (which served as the usual care comparator). Women were randomised to receive the intervention which comprised 25 home visits focusing on promoting child health and development via parenting and the home environment, from pregnancy to child age 2 years.

Within the right@home trial, mothers and children were followed-up with comprehensive annual assessments until children turned 6. Mothers provided written informed consent to re-enrol in extended follow-up at child age 2 years (n = 558/722 (77%)) and 5 years (n = 406/558 (56%)). The 6-year assessments began in May 2019 and were underway when COVID-19 was confirmed in Australia and lockdown measures were enacted. From 6 May to 6 December 2020, we assessed families' experiences of the COVID-19 pandemic and restrictions as well as mothers' and children's mental health, as part of the 6-year assessment (children were aged 5.9–6.4 years). For those who had already completed the 6-year assessment before 6 May 2020, we invited them to complete a stand-alone COVID-19 survey (children were aged 6.1–7.2 years).

The hypothesised model of the clinical, financial and social impacts of COVID-19 and their associations with maternal and child mental health are shown in Fig 2. For the current study,

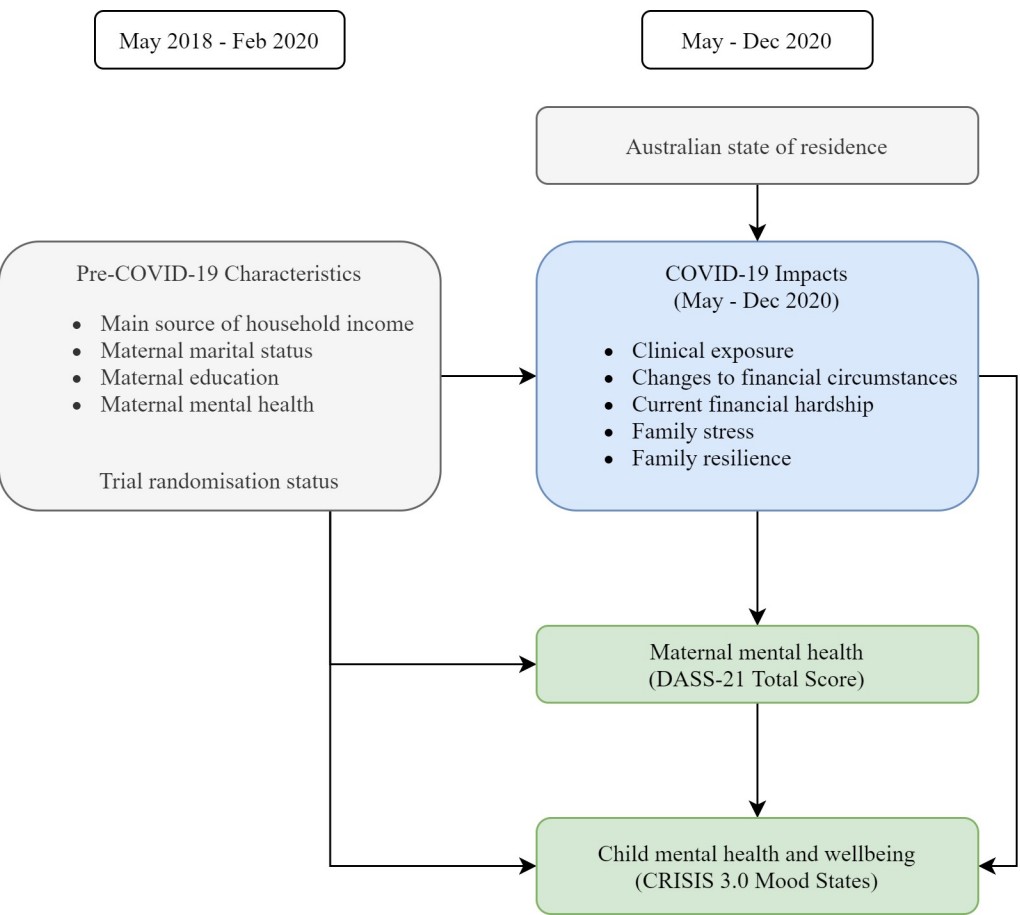

**Fig 2. Conceptual model of COVID-19 impacts and their associations with maternal and child mental health.**

all mothers and children who participated in the COVID-19 follow-up (both intervention and usual care arms) were included as a single observational cohort; randomisation status was controlled for in all analyses to account for the original trial study design.

## Participants

The right@home trial recruited pregnant women attending the antenatal clinics of 10 public maternity hospitals in metropolitan and regional areas of Victoria and Tasmania, Australia, between 30 April 2013 to 29 August 2014. Inclusion criteria were women with (i) expected due dates before 1 October 2014, (ii) less than 37 weeks gestation into their pregnancy at the time of recruitment, (iii) sufficient English proficiency to complete assessments, (iv) home addresses within travel boundaries of the study and (v) self-reported two or more of 10 antenatal adversity risk factors including: young pregnancy; not living with another adult; no support in pregnancy; poorer health; a long-term illness, health problem or disability that limits daily activities; coping difficulties; low education; no person in the household who currently earns an income; and never having had a job before [33, 34]. Antenatal risk factors were selected to identify mothers who were at risk of experiencing ongoing adversities known to be associated with poorer child health and development, and who may benefit from additional support for themselves and their child. Exclusion criteria were women who (i) were enrolled in an existing Tasmanian nurse home visiting program, (ii) did not comprehend the recruitment invitation

(e.g. had an intellectual disability such that they were unable to consent to participation, or had insufficient English to complete assessments), (iii) had no way to be contacted, or (iv) experienced a critical event that excluded their participation (termination of pregnancy, still birth, participant or child death).

## Measures

**COVID-19 impacts.** COVID-19 impacts were assessed using measures of clinical exposure, changes to financial circumstances, current financial hardship, family stress and family resilience. Items were drawn from the Coronavirus Health and Impact Survey Version 3 (CRISIS 3.0) [35, 36], the Australian Temperament Project Generation 3 (ATPG3) [37] and the Household, Income and Labour Dynamics in Australia (HILDA) Survey [38]. Details of individual items and the derived exposure summary measures for each COVID-19 impact are shown in Table 1. The Australian state in which the mother lived at the time of the COVID-19 data collection was also recorded.

**Maternal mental health.** Maternal mental health was measured using the 21-item short form of the Depression Anxiety and Stress Scales (DASS-21) [39]. The DASS-21 was selected as it is a widely used and validated measure of mental health [40] and is consistent with the repeated measurement throughout the right@home trial. Items are rated on a 4-point scale ("not at all" to "most of the time") assessing symptoms of depression, anxiety and stress experienced during the past week. Scores are summed to produce a continuous total score (range 0–63). Higher scores indicate poorer mental health.

**Child mental health and wellbeing.** Child mental health and wellbeing was measured using the 8-item parent-reported CoRonavIruS Health Impact Survey 3.0 Child Mood States scale (CRISIS 3.0) [35, 36]. The CRISIS 3.0 Child Mood States scale was selected as a newly validated but widely adopted measure of child mental health in the context of the COVID-19 pandemic. Items are rated on a 5-point scale, indicating the extent to which the child has experienced feeling worried, anxious, sad and depressed, tired, distracted or irritable and angry over the past 2 weeks. Scores are calculated as the mean of all 8 items, to produce a single scale score (range 0–5). Higher scores indicate more negative mood.

**Pre-COVID-19 characteristics.** Pre-COVID-19 characteristics were included as potential confounders when examining the association between COVID-19 impacts and mental health. Data were drawn from the routine follow-up completed most recently, prior to substantial COVID-19 occurrence in Australia (defined as 1 March 2020). This was either the 5-, 5.5- or 6-year follow-up (each being within the year preceding the COVID-19 data collection). Items included the main source of household income (government benefit or pension versus paid employment); mother's marital status (single or not living with a partner versus married or living with a partner); mother's education (did not versus did complete high school or any further training); and maternal mental health assessed using the DASS-21.

## Statistical analyses

Clinical, financial and social impacts of COVID-19 were described (n, %) for the whole cohort. Supplementary analyses stratified by state (Victoria, Tasmania, Other) and stage within the COVID-19 restrictions (6 May-1 June, 2 June-7 July, 8 July-23 Nov) were conducted to investigate whether experiences varied across the different states and different levels of restrictions.

Associations between COVID-19 impacts and maternal mental health (DASS Total Score) and child mental health (CRISIS Child Mood States scale) were estimated using linear regression models. The COVID-19 impacts (Table 1) were examined as exposure measures of: clinical exposure (mother or child self-quarantine); changes to financial circumstances (mother

**Table 1. Measures of families' experiences during the COVID-19 pandemic and public health restrictions (COVID-19 impacts).**

| | Items | Summary measure |
|---|---|---|
| Clinical exposure | Four items assessing mother, child, household family, and non-household family members who had a positive (confirmed case) test result ('yes' / 'no'); two items assessing mother and child who had a negative test result ('yes' / 'no'); and two items assessing whether mother or child had been required to self-quarantine ('yes'/'no'). Items drawn from the CRISIS 3.0 tool [35, 36].<br>Presenting for free COVID-19 testing and undertaking self-quarantine was required under Australian guidelines for returned travellers, confirmed cases, close contacts of confirmed cases, those with symptoms, and those awaiting test results. | Given the overall low incidence of cases in Australia, it was anticipated that there would be few confirmed cases within this cohort. Thus, mother or child experience of self-quarantine were used to create a single item reflecting any self- quarantine, as the summary measure of clinical exposure. Mother and/or child self-quarantine: 'yes' / 'no'. |
| Changes to financial circumstances | Two items assessing mother and family member lost job ('yes'/'no') and two items assessing mother and family member had reduced ability to earn ('yes'/'no'). Items drawn from the CRISIS 3.0 tool [35, 36]. | Mother experienced job loss and/or reduced ability to earn: 'yes'/'no' |
| Current financial hardship | Five items assessing family experience of the following financial hardships/difficulties paying: mortgage or rent; household bills; food; healthcare; home or car insurance. All 'yes'/'no' items, drawn from the Household, Income and Labour Dynamics in Australia (HILDA) Survey Wave 18 Household Questionnaire Material Deprivation Module [38]. | Current financial hardship: total count of five financial hardship items. Range 0–5, higher scores indicate more financial hardships. |
| Family stress/ negative change | Five items, rated on a 5-point scale: how worried mother is about self or family/friends being infected; how worried child is about self being infected; how worried child is about family/friends being infected; how stressful changes in contact with family have been for child; how difficult at home learning has made paid work/home duties for mother. Items drawn from the CRISIS 3.0 tool [35, 36]. | Family stress/negative change: Mean of five items. Range 0–5, higher scores indicate more stress and negative change. |
| Family resilience/ positive change | Seven items, rated on a 5-point scale: overall rating of quality of relationships with family and friends; how focused and productive mother has felt in work/domestic duties; overall positive changes in family and community; rating of quality of relationships between child and siblings; extent to which family has found good ways of coping; household/family have provided support to people in the community; child is enjoying online learning. Range 0–5, higher scores indicate more positive change and resilience. Items drawn from the CRISIS 3.0 tool [35, 36], and the ATPG3. | Family resilience/positive change: Mean of seven items. Range 0–5, higher scores indicate more resilience and positive change. |

lost job or had reduced ability to earn); total financial hardship; family stress; and family resilience. Both maternal and child mental health outcomes were rescaled to z-scores (mean 0, standard deviation 1) to standardise scores and assist interpretation.

Linear regression models examined COVID-19 impacts for associations with maternal and child mental health. Exposures (COVID-19 impacts) were first examined individually to

identify evidence of associations for each COVID-19 impact with maternal and child mental health, adjusting for pre-COVID-19 characteristics of household income, mother's marital status and mother's education; State of residence at the time of COVID-19 data collection (Vic, Tas or Other); and randomisation status (intervention versus control) (Model 1). A second series of models additionally adjusted for pre-COVID-19 maternal mental health (Model 2). Finally, all exposures (COVID-19 impacts) were analysed in a single model to identify evidence of their unique associations with maternal and child mental health, after accounting for each of the other exposures. Model 3 also adjusted for all potential covariates included in Model 2. We anticipated that there would be co-occurrence between each of the COVID-19 impacts and pre-COVID-19 maternal mental health; we therefore present all models to allow interpretation of these associations at each level of adjustment.

Data were analysed using Stata 16.0 for Windows (Stata Corp, College Station, TX).

### Ethical approval

The right@home trial, including the COVID-19 follow-up, were approved by the Royal Children's Hospital Human Research Ethics Committee (HREC 32296), Australia.

### Results

Of the 406 women enrolled in the extended 5- to 8-year follow-up, 319 (79%) completed the COVID-19 data collection either as part of their routine 6-year follow-up (n = 123) or as a stand-alone survey (n = 196) (Fig 3). Participant characteristics, including pre-COVID-19 characteristics, are shown in Table 2. Children were aged between 5.9 and 7.2 years at the time of the COVID-19 survey (mean 6.4 years, SD 0.3). The majority of families were living in Victoria (n = 210, 66%) or Tasmania (n = 97, 30%) at the time of the survey. Pre-COVID-19 characteristics showed mothers were experiencing high levels of adversity in the year before the pandemic: 53% were not in paid employment, 41% received their main source of household income from a benefit or pension, 21% had not completed high school or any further education and 27% were not living with another adult.

Table 3 shows rates of clinical exposure, changes to financial circumstances and current financial hardship. Within this cohort, one family reported the mother and the child had received a positive test result (confirmed case) for COVID-19, while 7% (n = 22) reported a household family member and 3% (n = 8) reported a non-household family member had a positive test result. Self-quarantine of either the mother or child was reported by 20% (n = 63). Regarding changes in financial circumstances, 27% of mothers (n = 85) reported job or income loss for themselves and 27% (n = 83) reported job or income loss for a family member; 44% (n = 139) in total reported any job or income loss. These figures were similar across State of residence and timing of data collection relative to the level of lockdown in place, although rates of self-quarantine were higher amongst those living in Victoria and those assessed between 8 Jul and 6 Dec, which aligns with Metropolitan Melbourne's second wave of infections and stringent lockdown (S1 Table).

Maternal reported COVID-19 impacts of family stress and resilience are shown in Fig 4. One third of mothers reported being moderately to extremely worried about themselves or family members becoming infected (30%) and that changes in contact with family and friends had been moderately to extremely stressful for their child (33%). Half of mothers (49%) reported that their child's at-home learning often or almost always made their usual duties more difficult. There were also positive changes; 48% of mothers reported that their family had often or almost always found good ways of coping and 29% reported providing support to people in the community.

Associations between COVID-19 impacts (clinical exposure, changes to financial circum-stances, financial hardship, family stress, and family resilience) and mental health for mothers and children are shown in Table 4. Regression coefficients (β) are presented as standardised coefficients, which can be interpreted as the number of standard deviations difference in the mental health outcome (comparable to effect sizes). The R-squared statistic ($R^2$) is also shown as the proportion of the variance in maternal and child mental health that is explained by each model. When examining each COVID-19 impact as individual exposures in separate

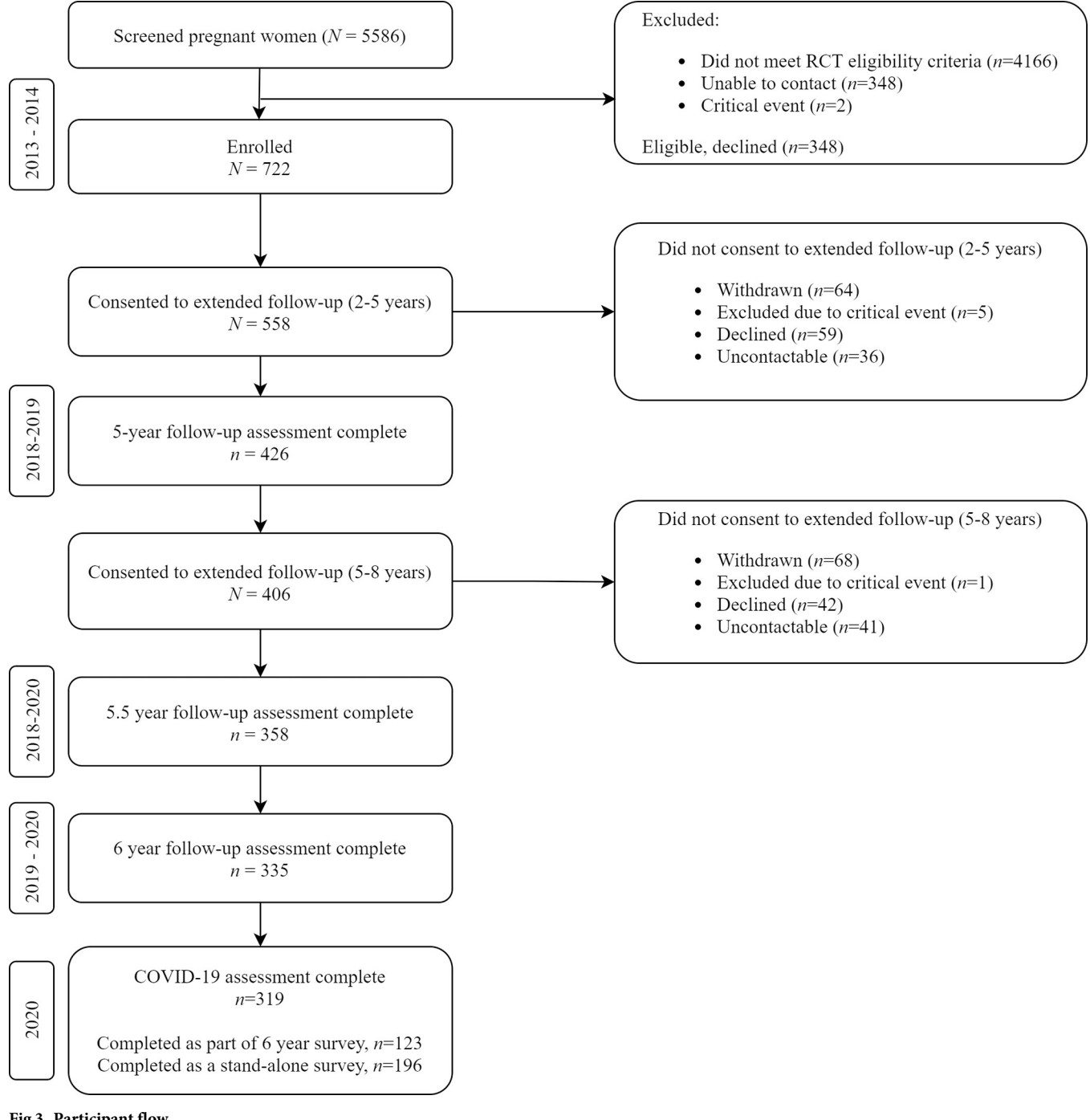

**Fig 3. Participant flow.**

**Table 2. Participant characteristics.**

|  | N % |
| --- | --- |
|  | N = 319 |
| **Child characteristics** |  |
| Gender (female) | 162 (50.8) |
| Age at assessment, years (mean, (SD) [range]) | 6.4 (0.3) [5.9–7.2] |
| **Randomisation Status** |  |
| Program Group | 168 (52.7) |
| Usual Care Group | 151 (47.3) |
| **Maternal Pre-COVID-19 characteristics** |  |
| Household income from benefit or pension | 129 (40.7) |
| Single or not living with partner | 121 (38.2) |
| Not living with another adult | 82 (26.5) |
| Did not complete high school or further education | 61 (20.8) |
| No paid employment | 167 (52.7) |
| **State of residence at COVID-19 follow-up** |  |
| Victoria | 210 (65.8) |
| Tasmania | 97 (30.4) |
| Queensland | 8 (2.5) |
| New South Wales | 3 (1.0) |
| South Australia | 1 (0.3) |

regression models, any self-quarantine ($\beta$ = 0.46, p = 0.003), greater financial hardship ($\beta$ = 0.27, p<0.001) and more family stress ($\beta$ = 0.49, p<0.001) were all associated with higher self-reported maternal mental health symptoms, while greater family resilience was associated with lower maternal mental health symptoms ($\beta$ = -0.40, p<0.001). These associations each accounted for pre-COVID-19 characteristics, state of residence and randomisation status (Model 1). The variance explained by each of these models ranged from 5% (any self-quarantine) to 13% (family stress). Associations for self-quarantine, greater financial hardship and family stress were also evident after additionally controlling for pre-COVID-19 maternal mental health, while the association for family resilience was attenuated (Model 2). Similar associations were identified for child mental health, although maternal job or income loss was associated with poorer child mental health ($\beta$ = 0.37, p = 0.006) while overall financial hardship was not (Model 1). The variance explained by each of these models ranged from 7% (maternal job or income loss) to 31% (family stress). The associations for child mental health were all still evident after additionally controlling for pre-COVID-19 maternal mental health (Model 2).

When all COVID-19 impacts were examined in a single model (Model 3), accounting for any co-occurring effects of the exposures, greater family stress showed the strongest association with both maternal ($\beta$ = 0.31, p<0.001) and child ($\beta$ = 0.68, p<0.001) mental health, and greater family resilience was associated with lower child mental health symptoms ($\beta$ = -0.36, p<0.001). Other associations were attenuated slightly. The variance explained by the models when accounting for all COVID-19 impacts as well as pre-COVID characteristics state of residence, randomisation status and pre-COVID-19 maternal mental health was 42% for maternal mental health and 39% for child mental health.

## Discussion

This study aimed to examine the clinical, financial and social impacts of the COVID-19 pandemic and their associations with maternal and child mental health, in a unique cohort of

**Table 3. Summary data of COVID-19 impacts (individual items and summary measures) and mental health outcomes.**

| | N% |
|---|---|
| | **N = 319** |
| **Clinical exposure** | |
| Mother–positive test result | 1 (0.3) |
| Mother–negative test result | 37 (11.6) |
| Mother self-quarantined | 56 (17.6) |
| Child–positive test result | 1 (0.3) |
| Child–negative test result | 21 (6.7) |
| Child self-quarantined | 36 (11.4) |
| Household family–positive test result | 23 (7.2) |
| Non-household family–positive test result | 8 (2.5) |
| *Mother/child self-quarantine summary* | *63 (19.8)* |
| **Changes to financial circumstances** | |
| Mother reduced ability to earn | 65 (20.4) |
| Mother lost job | 22 (6.9) |
| Family member reduced ability to earn | 51 (16.0) |
| Family member lost job | 36 (11.3) |
| *Mother lost job/income summary* | *85 (26.7)* |
| **Current financial hardship** | |
| Financial difficulties–mortgage, rent, loan repayments | 22 (6.9) |
| Financial difficulties–household bills | 44 (13.8) |
| Financial difficulties–food | 19 (6.0) |
| Financial difficulties–healthcare | 10 (3.1) |
| Financial difficulties–home, car insurance | 20 (6.3) |
| *Total financial hardship summary* (mean, (SD) [range]) | *0.4 (0.9) [0–5]* |
| **Family stress and resilience** | |
| *Family stress summary* (mean, (SD) [range]) | *2.1 (0.7) [1.0–4.8]* |
| *Family resilience summary* (mean, (SD) [range]) | *3.3 (0.6) [1.7–4.9]* |
| **Mental health outcomes** | |
| Maternal mental health (mean, (SD) [range]) | 9.6 (9.5) [0–55] |
| Child mental health (mean, (SD) [range]) | 2.3 (0.7) [1.0–4.4] |

Australian mothers recruited for their experience of adversity. In this cohort, only one mother and her child had experienced a COVID-19 infection. In contrast, high proportions of mothers reported negative financial and social impacts including job and income loss, worries about becoming infected, stress related to changes in contact with family and friends, and difficulties managing usual duties in addition to children's at-home learning. Many mothers also reported positive social impacts such as their family finding good ways of coping and supporting others in the community. Of these, both financial impacts and greater family stress were associated with poorer maternal and child mental health, while greater family resilience was associated with better mental health.

These results align with other data on Australian families. An intergenerational cohort surveyed between May to September 2020 found that 24% of parents had experienced job or income loss, [17] and a nationally-representative poll of Australian families in June 2020 reported a rate of 28% [6]. These are similar to the 27% of women reporting job and income loss in the current study. Like other studies from both Australian [6, 41, 42] and international cohorts [7, 28, 29, 32], we found that financial impacts of the COVID-19 pandemic and public

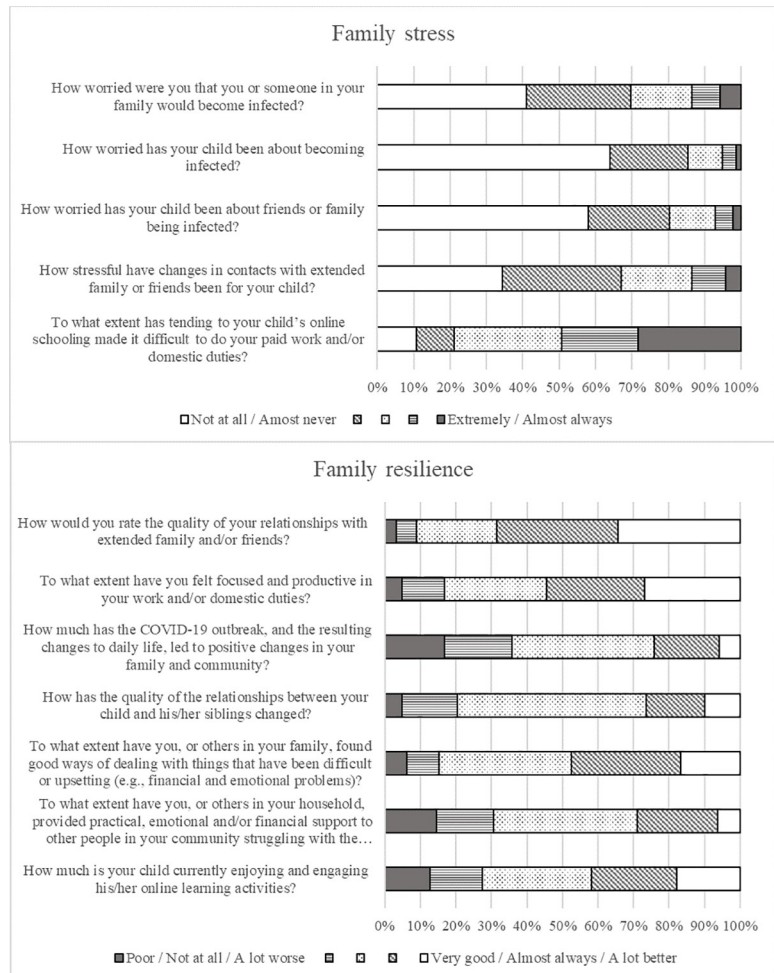

**Fig 4. Maternal report of family stress and family resilience individual items.**

health restrictions were associated with poorer parent and child mental health. For example, the Born in Bradford study of families living in an economically deprived city of the UK found more than one third of families experienced financial insecurity during the COVID-19 pandemic, and this was associated with parent anxiety and depression [7]. Our findings extend on the existing literature by demonstrating the relationship for Australian families experiencing adversity. Notably, families in the current cohort were already less likely to be in paid employment (53%) or have poorer mental health (20–30% with high mental health symptoms, see Bryson et al. [43]) before the pandemic. As such, the relative impacts of the public health restrictions are arguably greater for these families and likely to exacerbate existing health disparities.

Our findings highlight the critical importance of financial stability for families during crises such as the COVID-19 pandemic. Amongst families already at risk of financial hardship, rates of job and income loss were high. These findings were in the context of the Australian Governments' Job Keeper and Job Seeker policies (Fig 1), which mitigated some of the financial impacts for families. These responses represent some of the largest (albeit temporary) social policy changes in Australia's history. The JobKeeper and JobSeeker supplements were so significant that, by September 2020, overall levels of poverty and housing stress were substantially

**Table 4. Associations between COVID-19 impacts and maternal and child mental health outcomes.**

| | Model 1[†] | | | | Model 2[*] | | | | Model 3[*] | | | |
|---|---|---|---|---|---|---|---|---|---|---|---|---|
| | β[§] | 95% CI | p-value | $R^2$ | β[§] | 95% CI | p-value | $R^2$ | β[§] | 95% CI | p-value | $R^2$ |
| **Maternal mental health[‡]** | | | | | | | | | | | | 0.42 |
| Mother/child self-quarantine | 0.46 | 0.16 to 0.76 | 0.003 | 0.05 | 0.39 | 0.15 to 0.64 | 0.002 | 0.35 | 0.27 | 0.03 to 0.50 | 0.03 | |
| Mother lost job/income | 0.09 | -0.18 to 0.37 | 0.51 | 0.02 | 0.14 | -0.08 to 0.36 | 0.22 | 0.33 | -0.03 | -0.24 to 0.19 | 0.82 | |
| Total financial hardship | 0.27 | 0.14 to 0.40 | <0.001 | 0.07 | 0.20 | 0.09 to 0.31 | <0.001 | 0.36 | 0.14 | 0.04 to 0.25 | 0.009 | |
| Family stress | 0.49 | 0.33 to 0.65 | <0.001 | 0.13 | 0.37 | 0.23 to 0.50 | <0.001 | 0.38 | 0.31 | 0.17 to 0.45 | <0.001 | |
| Family resilience | -0.40 | -0.60 to -0.19 | <0.001 | 0.07 | -0.12 | -0.30 to 0.06 | 0.20 | 0.33 | -0.09 | -0.26 to 0.081 | 0.28 | |
| **Child mental health/wellbeing[‡]** | | | | | | | | | | | | 0.39 |
| Mother/child self-quarantine | 0.46 | 0.17 to 0.75 | 0.002 | 0.08 | 0.44 | 0.15 to 0.72 | 0.003 | 0.13 | 0.19 | -0.05 to 0.43 | 0.12 | |
| Mother lost job/income | 0.37 | 0.11 to 0.63 | 0.006 | 0.07 | 0.39 | 0.13 to 0.64 | 0.003 | 0.13 | 0.17 | -0.05 to 0.40 | 0.13 | |
| Total financial hardship | 0.05 | -0.08 to 0.19 | 0.43 | 0.05 | 0.02 | -0.11 to 0.15 | 0.81 | 0.10 | -0.10 | -0.21 to 0.01 | 0.08 | |
| Family stress | 0.74 | 0.60 to 0.88 | <0.001 | 0.31 | 0.70 | 0.57 to 0.85 | <0.001 | 0.34 | 0.68 | 0.54 to 0.82 | <0.001 | |
| Family resilience | -0.46 | -0.66 to -0.26 | <0.001 | 0.11 | -0.37 | -0.57 to -0.16 | 0.001 | 0.14 | -0.36 | -0.53 to -0.18 | <0.001 | |

Models 1 and 2 examined each COVID-19 impact as individual exposures in separate regression models; Model 3 examined all COVID-19 impacts together in a single model.

† Model 1 was adjusted for pre-COVID family income (paid employment vs benefit or pension), marital status (married/living with partner vs not), maternal education (did not vs did complete high school or any further education); state at the time of COVID-19 data collection (Vic, Tas or Other); and randomisation status (intervention vs control).

* Models 2 and 3 were adjusted for all of the above, and pre-COVID-19 maternal mental health.

‡ Outcome measures are rescaled to z-scores (mean 0, SD 1) to assist interpretation.

§ Regression coefficients (β) are standardised coefficients, comparable to effect sizes.

lower than they had been pre-COVID-19 [15]. However, the current study and wider Australian evidence show that many families were still financially impacted, [6, 17] and those families have fared poorly with regard to their mental health and wellbeing [42, 44]. The Jobseeker supplement has since been reduced and subsequently ceased entirely, leaving families at heightened risk of worsening economic positions and subsequent poorer mental health.

We further found that families reported high rates of stress related to changes in family interactions and worries about infection which were associated with poorer parent and child mental health, in line with previous studies [5, 7, 28, 29]. Previous findings include studies of families in the UK [7] Canada [29] and the US [30], where lockdown measures were put in place but COVID-19 infection rates were also high. Even with very low rates of infection in Australia, we identified similar findings in which an overall measure of family stress had the strongest associations with poorer maternal and child mental health. The most common of these experiences was maternal report that their child's online/at-home schooling made it difficult to do their usual paid work and/or domestic duties. Other Australian surveys found similar results; for example, parents who participated in the nationwide COVID-19 Pandemic Adjustment Survey in April 2020 reported that managing their child's at-home schooling was a significant challenge during the pandemic and related restrictions [44]. These findings are particularly pertinent to the large proportion of families in the current study living in Victoria, where at home-schooling was prolonged and access to at-school learning was limited for much of the 2020 school year (see Fig 1).

Promisingly, many families also reported experiences of family resilience. Mothers in the current study reported that their families found good ways of coping and had provided support to other people in the community; a total rating of these experiences of family resilience was associated with better mental health, particularly for children. Two nationwide surveys of

Australian families reported similar findings. Both the National Child Health Poll and the COVID-19 Pandemic Adjustment Survey found parents reported experiences of strengthened family relationships, constructive communication, spending more positive time together and feeling more connected with their children [6, 44].

The strengths of this study are the unique, prospective cohort of mothers with young children, who were recruited during pregnancy for their experience of adversity. Through the recruitment and retention processes of the trial, we have retained a cohort with experiences of adversity that are traditionally not well represented in longitudinal studies [45]. Even in the year prior to the COVID-19 pandemic, 5 to 6 years after recruitment, high proportions of this cohort were not in paid employment, reported low income and poor mental health. As this study was embedded in an established longitudinal study, we were able to quickly harness the opportunity to assess families' experiences during the height of the first lockdown and throughout the first 10 months of COVID-19 in Australia. We also had comprehensive pre-pandemic data to draw on, to account for family circumstances and mental health before the pandemic had emerged.

The study also has limitations. Despite the advantages of using an existing cohort, it was not designed *a priori* for assessing the impacts of the COVID-19 pandemic. The study includes a cohort of participants who were already engaged in a randomised trial of nurse home visiting and who had met the initial screening and inclusion criteria of the trial. Given that the families in this cohort have remained engaged in a longitudinal research project for a sustained period of 5 to 6 years, it is likely that the findings under-represent some of the families who were most socially vulnerable prior to and over the course of the pandemic. Hence the findings we describe may understate the full impact of the pandemic on families we couldn't represent, such as mothers without literacy in English who did not meet the eligibility criteria of the trial or those experiencing high levels of adversity who were lost to follow-up over the course of the trial. We also relied on maternal report for the family experiences and maternal and child mental health outcomes. These measures were necessarily maternal report, as mothers can provide valuable information about their families' experiences of the COVID-19 pandemic, as well as their young child's mental health. Maternal mental health is also an important determinant of child outcomes [46, 47]. However, mothers who experienced the pandemic as stressful, or those with poorer mental health, may be more likely to report more negative experiences and rate their child's mental health poorly [46]. To address this, analyses took account of self-reported maternal mental health data collected prior to substantial COVID-19 occurrence in Australia. Comparison between the models which did not (Model 1) and did (Model 2) account for prior maternal mental health showed that the findings and interpretation of results were not substantively changed, suggesting that the associations between COVID-19 impacts and mental health outcomes were somewhat independent from mothers' prior mental health.

Australian families clearly found the challenges of lockdown and the related changes in work-family life stressful, with flow-on effects for both parent and child mental health. Closure of schools and transitioning to at-home learning appears to be a key contributor to this. Studies from past instances of school closures suggest these actions disproportionately impact families experiencing adversity, by removing access to additional services provided at schools while increasing stress on primary caregivers [48]. As Australia (and countries globally) emerge from lockdown and the pandemic, the disproportionate burden on women and children already living in adversity will need to be addressed with similarly disproportionate service system and economic policy responses. Promisingly, findings of this and existing studies show that family resilience such as finding ways of coping together and supporting others were associated with better mental health; this suggests that there are opportunities to build positive community support into the recovery processes. Future research would benefit from

examining the pre-pandemic factors which might be associated with family resilience, to iden-
tify opportunities for supporting families both in recovering from the pandemic and in
redressing inequities in parent and child mental health.

## Conclusions

The financial and social impacts of Australia's public health restrictions have substantially
affected families experiencing adversity, and their mental health. This cohort of mothers and
children were already at disproportionate risk of poor mental health prior to the pandemic,
with now potentially worsening inequities as we have seen globally; even without the impact of
the virus itself. Unless the financial and social consequences of lockdown are addressed, the
inequities arising from adversity are likely to be exacerbated by this crisis. To recover from
COVID-19, the economic and healthcare needs of women and children living in adversity
must be prioritised. Policy investment in income support and universal and equitable access to
family health services are critical.

## Supporting information

**S1 Table. Summary data of COVID-19 impacts (individual items and summary measures)
and mental health outcomes, broken down by state and date of assessment.**
(DOCX)

## Acknowledgments

The "right@home" sustained nurse home visiting trial is a research collaboration between the
Australian Research Alliance for Children and Youth (ARACY); the Translational Research
and Social Innovation (TReSI) Group at Western Sydney University; and the Centre for Com-
munity Child Health (CCCH), which is a department of The Royal Children's Hospital and a
research group of Murdoch Children's Research Institute. We thank all families, the research-
ers, nurses and social care practitioners working on the right@home trial, the antenatal clinic
staff at participating hospitals who helped facilitate the research, and the Expert Reference
Group for their guidance in designing the trial.

## Author Contributions

**Conceptualization:** Hannah Bryson, Fiona Mensah, Anna Price, Lisa Gold, Penelope Dakin,
Tracey Bruce, Lynn Kemp, Sharon Goldfeld.

**Data curation:** Hannah Bryson, Bridget Kenny.

**Formal analysis:** Hannah Bryson, Fiona Mensah.

**Funding acquisition:** Anna Price, Penelope Dakin, Sharon Goldfeld.

**Investigation:** Hannah Bryson, Bridget Kenny.

**Methodology:** Hannah Bryson, Fiona Mensah, Anna Price, Lisa Gold, Shalika Bohingamu
Mudiyanselage, Penelope Dakin, Tracey Bruce, Kristy Noble, Lynn Kemp, Sharon
Goldfeld.

**Project administration:** Hannah Bryson, Bridget Kenny, Sharon Goldfeld.

**Supervision:** Sharon Goldfeld.

**Writing – original draft:** Hannah Bryson, Fiona Mensah, Anna Price, Lisa Gold, Shalika
Bohingamu Mudiyanselage, Bridget Kenny.

**Writing – review & editing:** Hannah Bryson, Fiona Mensah, Anna Price, Lisa Gold, Shalika Bohingamu Mudiyanselage, Bridget Kenny, Penelope Dakin, Tracey Bruce, Kristy Noble, Lynn Kemp, Sharon Goldfeld.

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
