## [Decision Letter · Decision Letter 0]

30 Jun 2021

PONE-D-21-19599

Clinical, financial and social impacts of COVID-19 and their associations with mental health for mothers and children experiencing adversity in Australia

PLOS ONE

Dear Dr. Goldfeld,

Thank you for submitting your manuscript to PLOS ONE. After careful consideration, we feel that it has merit but does not fully meet PLOS ONE’s publication criteria as it currently stands. Therefore, we invite you to submit a revised version of the manuscript that addresses the points raised during the review process.

We look forward to receiving your revised manuscript.

Kind regards,

Livio Provenzi

Academic Editor

PLOS ONE

Journal Requirements:

1. Please ensure that your manuscript meets PLOS ONE's style requirements, including those for file naming. The PLOS ONE style templates can be found athttps://journals.plos.org/plosone/s/file?id=wjVg/PLOSOne_formatting_sample_main_body.pdf and https://journals.plos.org/plosone/s/file?id=ba62/PLOSOne_formatting_sample_title_authors_affiliations.pdf

2. In ethics statement in the manuscript and in the online submission form, please provide additional information about the ethics approval for this study. Please clarify whether the initial ethics approval encompass this study or if there was specific approval for a COVID-specific study as part of the right@home study. If not, please confirm with your IRB that specific, prospective approval for the current study was not necessary and attach the correspondence as an 'Other' file for our records ('Other' files are not published). Thank you for your attention to this request."

patient records used in your retrospective study. Specifically, please ensure that you have discussed whether all data were fully anonymized before you accessed them and/or whether the IRB or ethics committee waived the requirement for informed consent. If patients provided informed written consent to have data from their medical records used in research, please include this information.

Additional Editor Comments (if provided):

Reviewers' comments:

Reviewer's Responses to Questions

**Comments to the Author**

1. Is the manuscript technically sound, and do the data support the conclusions?

Reviewer #1: Yes

Reviewer #2: Yes

2. Has the statistical analysis been performed appropriately and rigorously? 

Reviewer #1: Yes

Reviewer #2: Yes

3. Have the authors made all data underlying the findings in their manuscript fully available?

Reviewer #1: No

Reviewer #2: Yes

4. Is the manuscript presented in an intelligible fashion and written in standard English?

Reviewer #1: Yes

Reviewer #2: Yes

5. Review Comments to the Author

Reviewer #1: The present manuscript describes a research conducted on a sample of mothers who have had experience of adversities. The manuscript is well written, and the research seems sound and of interest (despite the limitations due to how the sample was recruited).

Nevertheless, I feel that there is some room for improvement and there are a few concerns that need to be addressed before it is suitable for publication.

1- The authors didn't make their data public: this might be in violation of PlosOne policies. The authors declare that this is due to data containing potentially re-identifiable and sensitive participant information. Given the peculiarities of the sample, this might be the case, and the authors explained very well the procedure to request access to the data.

2- Regarding the introduction, I don't have any particular comment, as it is well written and coherent. I have, however, some troubles with the pictures, as in the submitted files their quality is too poor. Fig1, in particular, is impossible to read, and the other figures are of very degraded quality. This needs to be amended prior to publication.

3- Regarding the methods, the study seems to be carefully designed and the methods are adequately described. There is a limitation regarding the sample, as this is a convenience sample recruited from a trial in 2013/14 with several inclusion criteria: this clearly impacts the generalizabilty of the results. The authors only partially address this in the discussion (p21 line unknown as line numbering disappeared from page 18...): I think the authors should state not only the limit itself, but also the potential impact it has.

4- Study measures are well described, and Table 1 is quite useful. However, may I suggest the authors to add in the supplementaries a copy of the survey? This would help replicability, and potentally allow the authors to remove some details from the text.

5- maternal mental health and child mental health were assessed on two different time-frames (respectively, past week and past 2 weeks). Why?

6- statistical analyses were performed, to the best of my knowledge, adequately. However, I don't understand why the authors decided to carry out and report 3 different models. What are model 1 and 2 adding to the results? If you hypothesise that all the measures of COVID-19 impact are contributing in determining you dependent variables, then only model 3 is relevant. Otherwise, the reason for this should be carefully explained in the manuscript.

7-regarding the regression models, the authors should also report the coefficient of determination (R2) for each model

8-in table 4 the cross near Model 1 has no footnote

In conclusion, this is an overall well written paper, describing a potentially interesting study. There are some issues with the figure quality and some minor changes to be done in the methods/results sections.

Reviewer #2: I thank you for the opportunity to review this manuscript which is very timely and falls within my area of interest. Analyses are well done and results have been clearly described.

I suggest to the authors some changes to improve the quality of their work.

In particular I would suggest author to expand - in the introduction section - the literature on the COVID-19 pandemic psychosocial impact on citizens in order to more widely discuss your results.

6. PLOS authors have the option to publish the peer review history of their article (what does this mean?). If published, this will include your full peer review and any attached files.

Reviewer #1: **Yes: **Lorenzo Palamenghi

Reviewer #2: No

---

## [Author Response · Author response to Decision Letter 0]

11 Aug 2021

We thank the Reviewers for their considered feedback. We have addressed the Reviewer comments in the attached document "Response to Reviewers".

---

## [Decision Letter · Decision Letter 1]

31 Aug 2021

Clinical, financial and social impacts of COVID-19 and their associations with mental health for mothers and children experiencing adversity in Australia

PONE-D-21-19599R1

Dear Dr. Goldfeld,

We’re pleased to inform you that your manuscript has been judged scientifically suitable for publication and will be formally accepted for publication once it meets all outstanding technical requirements.

Kind regards,

Livio Provenzi

Academic Editor

PLOS ONE

Additional Editor Comments (optional):

Reviewers' comments:

Reviewer's Responses to Questions

**Comments to the Author**

1. If the authors have adequately addressed your comments raised in a previous round of review and you feel that this manuscript is now acceptable for publication, you may indicate that here to bypass the “Comments to the Author” section, enter your conflict of interest statement in the “Confidential to Editor” section, and submit your "Accept" recommendation.

Reviewer #1: All comments have been addressed

Reviewer #2: All comments have been addressed

2. Is the manuscript technically sound, and do the data support the conclusions?

Reviewer #1: Yes

Reviewer #2: Yes

3. Has the statistical analysis been performed appropriately and rigorously? 

Reviewer #1: Yes

Reviewer #2: Yes

4. Have the authors made all data underlying the findings in their manuscript fully available?

Reviewer #1: No

Reviewer #2: Yes

5. Is the manuscript presented in an intelligible fashion and written in standard English?

Reviewer #1: Yes

Reviewer #2: Yes

6. Review Comments to the Author

Reviewer #1: (No Response)

Reviewer #2: (No Response)

7. PLOS authors have the option to publish the peer review history of their article (what does this mean?). If published, this will include your full peer review and any attached files.

Reviewer #1: **Yes: **Lorenzo Palamenghi

Reviewer #2: **Yes: **Serena Barello

---

## [Editor Report · Acceptance letter]

3 Sep 2021

PONE-D-21-19599R1 

Clinical, financial and social impacts of COVID-19 and their associations with mental health for mothers and children experiencing adversity in Australia 

Dear Dr. Goldfeld:

I'm pleased to inform you that your manuscript has been deemed suitable for publication in PLOS ONE. Congratulations! Your manuscript is now with our production department. 

Kind regards, 

on behalf of

Dr. Livio Provenzi 

Academic Editor

PLOS ONE